# Vitamin D Effects on the Immune System from Periconception through Pregnancy

**DOI:** 10.3390/nu12051432

**Published:** 2020-05-15

**Authors:** Bianca Schröder-Heurich, Clara Juliane Pacifica Springer, Frauke von Versen-Höynck

**Affiliations:** 1Gynaecology Research Unit, Hannover Medical School, Carl-Neuberg-Str.1, D-30625 Hannover, Germany; Schroeder-Heurich.Bianca@mh-hannover.de (B.S.-H.); Clara.J.P.Springer@stud.mh-hannover.de (C.J.P.S.); 2Department of Obstetrics and Gynaecology, Hannover Medical School, Carl-Neuberg-Str.1, D-30625 Hannover, Germany

**Keywords:** vitamin D, pregnancy, preeclampsia, immune system, assisted reproduction, reproductive tissue, vitamin D deficiency, recurrent pregnancy loss

## Abstract

Vitamin D is a well-known secosteroid and guardian of bone health and calcium homeostasis. Studies on its role in immunomodulatory functions have expanded its field in recent years. In addition to its impact on human physiology, vitamin D influences the differentiation and proliferation of immune system modulators, interleukin expression and antimicrobial responses. Furthermore, it has been shown that vitamin D is synthesized in female reproductive tissues and, by modulating the immune system, affects the periconception period and reproductive outcomes. B cells, T cells, macrophages and dendritic cells can all synthesize active vitamin D and are involved in processes which occur from fertilization, implantation and maintenance of pregnancy. Components of vitamin D synthesis are expressed in the ovary, decidua, endometrium and placenta. An inadequate vitamin D level has been associated with recurrent implantation failure and pregnancy loss and is associated with pregnancy-related disorders like preeclampsia. This paper reviews the most important data on immunomodulatory vitamin D effects in relation to the immune system from periconception to pregnancy and provides an insight into the possible consequences of vitamin D deficiency before and during pregnancy.

## 1. Introduction

While chemically different types of vitamin D exist, vitamin D_3_ is the most active and important form in humans with several important functions including its classical role in calcium homeostasis and bone health. In the body, the active form of vitamin D_3_ (1,25-(OH)_2_D; calcitriol) is synthesized in several steps. The precursor molecule cholecalciferol (vitamin D_3_) is formed in the skin by exposure to UV-B light, although a small amount of vitamin D_3_ is also obtained from food intake and is the major form of vitamin D in supplements (Figure 1). Cholecalciferol is further metabolized by several hydroxylation steps, first into calcidiol (25-(OH)D) which is reflective of body stores of vitamin D and then into the active form calcitriol (1,25-(OH)_2_D) [1], [2]. While the measurement of the active vitamin D_3_ is not recommended due to the short half-life time in the blood, serum 25-(OH)D is used to determine whether a patient’s vitamin D_3_ status is deficient, sufficient or intoxicated [3]. This precursor form correlates with the active form of vitamin D_3_ in the body. In most publications included in this review, a 25-(OH)D content of <20 ng/mL is defined as vitamin D deficient, 21–30 ng/mL as insufficient and over 30 ng/mL as sufficient. In the studies where the thresholds deviate from this, we have specified it further in the text. Serum 25-(OH)D concentrations can be given in the units nmol/L or ng/mL (to convert nmol/L to ng/mL, divide the value by 2.5). The term vitamin D is generally used in this review to refer to vitamin D_3_ as well as its metabolites, unless the vitamin D metabolites or precursors are distinguished for further clarification.

Vitamin D deficiency is a global problem that affects all ethnic groups, especially older people, children and pregnant women. During pregnancy an adequate supplementation of vitamin D is of great relevance, as vitamin D deficiency is associated with adverse pregnancy outcomes, e.g. recurrent pregnancy loss and a higher risk of preeclampsia [5,6]. In the following, we will review on the immune modulatory effects of vitamin D in female reproductive tissues and clinically relevant outcomes. 

## 2. Vitamin D metabolism and Synthesis in Reproductive Tissue

Over the past decades, many so far unknown actions of vitamin D have become apparent involving regulation of genes, endothelial function and integrity [7], and the immune system which includes to maintain tolerance and to promote protective immunity [6,8]. Vitamin D affects inflammatory responses by modulation of the expression of genes which generate pro-inflammatory mediators or the activation of signaling cascades which regulate inflammatory responses [9]. B-lymphocytes, T-lymphocytes, dendritic cells (DCs) and macrophages, which are all able to synthesize intracrine vitamin D are modulated in their proliferation and differentiation by vitamin D (Figure 1) [5]. The influence of vitamin D on the adaptive immune system involves inhibition of B cell proliferation and differentiation as well as inhibition of T cell proliferation, resulting in a shift towards T helper (Th) 2 of the Th1/Th2 immune balance [5,10,11,12,13,14]. Furthermore, vitamin D affects the T cell response by stimulating Th2 cytokines and reducing Th17 cytokines [15,16]. An effect on immune tolerance by vitamin D on the differentiation of regulatory T cells (Tregs) [17,18] and inhibition of maturation of dendritic cells (DC) has also been reported [19]. Additional effects of vitamin D on the innate immune system through the induction of antimicrobial peptides in macrophages and through an activation of 1α-hydroxylase, an enzyme which catalyzes the synthesis of vitamin D_,_ and the associated activation of Toll-like receptors (TLR) on macrophages have been observed [5]. Further, vitamin D showed an anti-inflammatory effect by inhibition of NFκB in in vitro studies [20,21].

By binding of 1,25-(OH)_2_D to its high affinity receptor (vitamin D receptor; VDR) the activation of numerous target genes occurs through transcription leading either to inhibiting or activating processes in the cell. VDR and CYP27B1 (1α-hydroxylase) are expressed in various tissues and cells and have also been found in female reproductive tissues such as ovary (granulosa and theca cells), endometrium, decidua and placenta [4,22,23,24,25,26] (Figure 2) which extended the immunomodulatory function of vitamin D to the maternal-fetal interface during pregnancy. An association between maternal vitamin D status and the prevalence of bacterial vaginosis in early pregnancy was found in a cohort of 469 pregnant women suggesting the immune system response to microbial invasion may be influenced by vitamin D (18). Decidual vitamin D could have an effect on antimicrobial reactions by modulating differentiation of macrophages [27]. In addition, the endometrial protein expression of CYP27B1 and VDR in endometrial stroma cells as well as in decidual cells was detected, whereby the CYP27B1 mRNA levels were induced by the pro-inflammatory cytokine IL1-β in endometrial cells [22]. In a different study of the endometrial cycle VDR was downregulated in the mid-secretory phase compared to the early secretory phase [28]. Controversial results were described by Bergada et al. who observed a decrease in VDR expression in the proliferative endometrium compared to the secretory phase in a cohort of 60 women [29]. Further, the placenta and decidua are able to produce 1,25-(OH)_2_D and 24,25-(OH)_2_D and components of the vitamin D signaling pathway like CYP27B1, CYP24A1 and VDR [30] (Figure 2). The expression of VDR on maternal decidua and fetal trophoblasts implies that 1,25-(OH)_2_D produced by the placenta acts in an autocrine or paracrine manner and exerts effects on both maternal and fetal cells [31]. Liu and colleagues investigated the anti-inflammatory effect of vitamin D in the placenta using in vitro and ex vivo mouse models. Dysregulation of vitamin D metabolism by CYP27B1 knockout and function by VDR down regulation lead to an aberrant inflammatory response in placental fetal cells, suggesting that vitamin D plays a regulatory role in the immunological response at the fetal-placental interface during pregnancy [31]. In a study of Chary et al. [32] an impairment of Tregs in pregnant women with vitamin D deficiency has been observed. 153 pregnant women with different 25-(OH)D status were assessed for regulatory T cells and IgE receptor (CD23 and CD21) expression on B cells showing that the Treg cell population in maternal and umbilical cord blood in the group of 25-(OH)D deficient pregnant women were lower compared to 25-(OH)D sufficient or insufficient pregnant women. B cells with CD21 and CD23 expression were higher in maternal blood of 25-(OH)D deficient pregnant women. In addition, TGF-β and IL-10 in maternal and cord blood were decreased in 25-(OH)D deficient and insufficient subjects. The regulatory T cell transcription factor (FOXP3) and VDR were downregulated in the placenta in the group of 25-(OH)D deficient women while CD21 and CD23 expression were increased in 25-(OH)D deficient and insufficient participants [32]. Recently it was shown that the inflammatory response to bacterial infection of the placenta is closely associated with vitamin D signaling. Neutrophiles and macrophages treated with a combination of bacterial lipopolysaccharide (LPS) and 1,25-(OH)_2_D showed phagocytic capacity [33]. Epidemiological studies on placental inflammation and vitamin D levels, however, have been inconclusive. Puthuraya and colleagues [34] analyzed levels of vitamin D on the first day postpartum in the serum of women who gave birth to very low birth weight infants and the incident of placental inflammation in the same women. In a logistic regression analysis, they did not find vitamin D deficiency and placental inflammation to be associated. Zhang and Chen et al. [33,35], on the other hand, showed that vitamin D levels (deficient (<27.5 nM), insufficient (49.99–27.5 nM), sufficient (≥ 50 nM)) were significantly lower in women with placental inflammation.

Furthermore, 1,25-(OH)_2_D has been shown to increase production of cathelicidin in keratinocytes, macrophages, neutrophiles [36] and placental decidua cells [37,38]. Cathelicidin is an antimicrobial peptide capable of inducing bacterial and cellular apoptosis. In an experiment conducted by Liu et al. [38] infection rates and cell death were reduced in trophoblasts in vitro, when pre-treated with 25-(OH)D, 1,25-(OH)_2_D or both. In vivo experiments showed that 62.5% of pregnant mice treated with LPS experienced early pregnancy loss and elevated levels of TNF-α, INF-γ and macrophage inflammatory protein 2 (MIP-2). However, a cohort pre-treated with vitamin D_3_ exhibited only 14.3% early pregnancy losses and less cytokine elevation [39]. Therefore, Yates et al. [40] suggest, it might be possible to inhibit placental inflammation by supplementing women with vitamin D in early pregnancy.

Taken together, these data show that vitamin D affects multiple arms of the immunological response in female reproductive tissues. 

## 3. The Effect of Vitamin D on the Immune System in the Periconception Period and in Pregnancy 

By now it is an established fact that the immune system is involved in the implantation process and pregnancy in general and takes on important roles in fertility, implantation and pregnancy. 

Directly post-coitus, the seminal fluid induces a pro-inflammatory immune response, activating neutrophils and macrophages [41], as well as cytokine and chemokine pathways [42]. This response contributes to the endometrial remodelling necessary for implantation. It has been suggested, that the maternal immune system acts as a quality control for semen. Less immunogenic seminal fluid leads to impaired endometrial receptivity [41]. Induced by the reaction to paternal semen, maternal Tregs are involved in building immune tolerance towards paternal antigens [43]. Sufficient TGF-β quantity in the seminal fluid supports the adequate Treg activation [41]. Effective tolerance is critical to the implantation of the embryo. In the oviduct sperm cells binding to the epithelium, stimulate an anti-inflammatory response by upregulating TGF-β and IL-10 and thereby supporting spermatocyte passage and oocyte fertilization [42]. In bovine oviductal fluid and secretions from bovine oviductal epithelial cells, Pillai et al. identified maternal immune factors [44]. TNF-α is believed to play a role in pre-implantation embryo transport by inducing local contractions. CSF1 (macrophage colony stimulating factor 1) in oviductal epithelial cells accelerates embryo development in cattle [45]. LIF (leukemia inhibiting factor) is known to positively impact oocyte fertilization and early embryo development in sheep and cattle [46]. LIF and CSF were also induced by seminal fluid in mice [47], while IL-8 promotes mitosis in endometrial cells [48]. 

After implantation, tolerance is upheld through repression of cytotoxic T cells, Th1 cells, macrophages, DCs and NK cells by Tregs. Effector functions of Tregs are promoted by 1,25(OH)_2_D [49] which have immuno-suppressant functions and are of critical importance to the establishment of pregnancy. In endometrial stem cells 1,25(OH)_2_D reduces most cytokine production, like IL-6, which blocks Treg development, but up-regulates TGF-β, which is capable of activating Tregs [50,51]. Further, 1,25(OH)_2_D has been found to promote DCs with tolerogenic properties by inhibiting their maturation. In tolerogenic DCs 1,25(OH)_2_D reduces the production of Th1-activating IL-12 and increases the production of the Th2 cytokine IL-10 [52]. These DCs also play a major role in the activation of Treg [50,53]. It has been postulated, that the immuno-suppressant effect of Tregs under the influence of DCs minimizes the obligatory inflammation following implantation in order to prevent resorption of the early embryo [43]. In mice, the depletion of Tregs using an anti-CD25 antibody lead to implantation failure and fetus rejection [54]. 

The adaptive immune system regulates the maternal immune tolerance towards the fetus during pregnancy. The correct balance of Th1 cytokines like TNF-α, INF-γ and IL-2 and Th2 cytokines such as IL-4, IL-5, IL-6, IL-9, IL-10 and IL-13 [53] is of great importance for a healthy pregnancy. The predominance of Th2 cells and humoral immunity is generally associated with normal pregnancy [43]. 1,25-(OH)_2_D has been shown to selectively inhibit Th1 cells and to enhance Th2 differentiation by a direct influence on CD4^+^ progenitor cells [55]. By reducing Th1 cytokines and promoting Th2 cytokines [53,55,56], 1,25-(OH)_2_D makes the maternal immune system especially sensitive to pathogens, while diminishing self-destructive mechanisms of effector T cell subsets [57,58]. Ikemoto et al. [59] recently investigated the effect of vitamin D on the Th1/Th2 balance in infertile women. They found that more than 80% of the tested infertile women had vitamin D insufficiency or deficiency, nearly half of which showed increased Th1/Th2 ratios. Interestingly, through supplementation of vitamin D, the Th1/Th2 ratio was lowered significantly.

Within the innate immune system, uterine NK (uNK) cells have also been shown to respond to vitamin D regulation. uNK cells are involved in the regulation of spiral arterial remodeling and trophoblast invasion and are therefore critical to a successful implantation. In general, 1,25(OH)_2_D induces uNK cell activation [60]. After 1,25-(OH)_2_D and 25-(OH)D treatment for 28 h, first-trimester uNK cells synthesized less granulocyte-macrophage colony stimulating factor 2, TNF-α and IL-6 and additionally expressed more TLR4 [37], thereby enhancing pathogen recognition and reducing inflammation. An overview of in vitro, in vitro, observational and interventional studies on immunomodulatory effects of vitamin D in reproductive tissues and reproductive outcomes is given in Table 1.

## 4. Vitamin D, the Immune System and Adverse Reproductive Outcomes 

### 4.1. Recurrent Pregnancy Loss

Recurrent pregnancy loss (RPL) affects 1–2% of reproductive women. The exact prevalence is hard to estimate as it depends on the definition used [76]. According to the most recent guidelines of the European Society of Human Reproduction and Embryology (ESHRE) [76] and the American Society of Reproductive Medicine (ASRM) [77] RPL is defined as the loss of 2 or more pregnancies. However, the World Health Organization (WHO) and professional associations of several countries follow a definition of 3 or more consecutive losses before 20 weeks gestation in their guidelines [78,79].

While in 50% of women affected by RPL the underlying cause cannot be determined more frequent pathogenic mechanisms include uterine anomalies, genetic anomalies, metabolic and endocrine disruptions and immunologic factors.

Activation of the immune system leads to an unfavorable situation for implantation and increased probability of RPL [80,81,82,83,84]. In 20% of affected women abnormalities of the cellular immune system and autoimmune conditions, e.g. antiphospholipid antibodies (aPL) can be detected. Abnormalities include increased peripheral and uNK cell levels, NK cell cytotoxicity, Th 1/Th2 ratios, Treg/Th17 ratios and higher secretion of cytokines, e.g. TNF-α [69,72,85,86,87,88].

Recent studies suggest an association between vitamin D deficiency and auto- or allo-immunologic disruption in RPL. However, most studies are observational and of the available clinical studies the minorities are randomized controlled trials (RCT) as presented in a comprehensive review by Goncalves et al. [89].

In a study by Ota et al. 47% of patients with RPL were vitamin D deficient (<30 ng/mL, *n* = 63) [67]. In those women a higher rate of APL-, ANA-, and thyroid peroxidase (TPO)-antibodies and higher numbers of CD19+B- lymphocytes and NK cells were found compared to women with normal vitamin D status (≥ 30 ng/mL, *n* = 70). Cytotoxicity of these cells was reduced by 1,25-(OH)_2_D in vitro [62,67]. Similar results were reported by Chen et al. who determined the effect of 1,25-(OH)_2_D on the number of peripheral blood cells, Th1 cytokines, and NK cytotoxicity in 99 women with RPL [71]. The percentage of CD19+ B-cells and NK cytotoxicity and the proportion of TNF-α-expressing Th cells were significantly higher in the vitamin D insufficient group than in the group with normal vitamin D levels. After supplementation of 0.5 μg/day of vitamin D for 2 months the percentage of CD19+ B-cells and of TNF-α-producing Th cells as well as NK cytotoxicity was significantly lower after treatment when compared with before treatment. In an observational study by Rafiee et al. a decline in the Th17 frequency and Treg cells and in the ratio of Th17/Treg in women who were treated with lymphocyte immune therapy was observed [73]. The decrease was significantly more in the study arm which additionally received vitamin D. In one RCT vitamin D supplementation of 0.25 μg daily starting ≤ 6 weeks gestation was associated with a significant reduction of IFN-γ levels and an increase of successful pregnancies. However, the results were not statistically significant most probably due to small sample size [74]. In the second RCT 77 pregnant women with a history of RPL and similar vitamin D and IL-23 levels at study start were assigned to 2 groups [75]. While the study group received oral vitamin D (400 IU/d daily) and vaginal progesterone (400 mg daily), the control arm received placebo tablets and vaginal progesterone (400 mg daily). IL-23 levels decreased in the study group and increased in the control group and IL-23 and vitamin D showed an inverse relationship. However, while the incidence of RPL was less in the study arm the results were not significantly different from the control arm when confounding factors were additionally considered.

Some studies investigated whether vitamin D status and exposure impacts immunologic aspects of the endometrium and the maternal-fetal interface. Whole endometrial cells from women with a history of RPL (*n* = 8) secreted significant higher amounts of IFN-γ compared to women with at least 1 healthy life birth without spontaneous abortions or infertility (*n* = 8) [50]. After 1,25(OH)_2_D_3_ exposure from women with RPL produced significantly less IFN-γ. Both groups converted 25-(OH)D to active vitamin D suggesting a comparable capacity of the endometrium to produce or respond to vitamin D in RPL. Also, expression of VDR, CYP27B1 and CYP24A1 was similar between women with RPL and the control group [90].

In contrast VDR and CYP27B1 expression levels were reduced in chorionic villi and decidua in RPL compared to gestational age matched women with voluntary pregnancy termination [66,91]. While in the RPL group serum 25-(OH)D concentrations were also reduced they might correlate with VDR expression levels at the maternal-fetal interface and contribute to poorer outcomes. In RPL levels of the anti-inflammatory cytokine IL-10 were significantly reduced in chorionic villi and decidua while inflammatory cytokine levels (TNF-α, IL-2, IFN-γ) were markedly increased compared to the control group [66]. Similar results were obtained by Li et al. in decidual tissue who reported lower 25-(OH)D and TGF-β concentrations, lower VDR expression and higher concentrations of IL-17 and IL-23 in RPL cases [68].

The available data suggest that low vitamin D concentrations could be a contributor to immunologic alterations in RPL and one may speculate that vitamin D_3_ might be a therapeutic option in this specific group of women. Although immunologic benefits of vitamin D were reported in several observational studies the 2 small randomized controlled trials failed to show a significant correlation between vitamin D supplementation and the incidence of RPL. Therefore, further randomized clinical studies with sufficient numbers of participants at best starting preconception are required to investigate the association between vitamin D deficiency, supplementation and RPL.

### 4.2. Recurrent Implantation Failure

Recurrent implantation failure (RIF) is diagnosed when good-quality embryos repeatedly fail to implant after transfer in several IVF treatment cycles. The main pathological factors of recurrent implantation failure involve a) the blastocyst, b) the endometrium and c) a combination of a) and b). Investigations of endometrial biopsies in RIF patients revealed maturation defects of the endometrium, a disbalance of immune cells and a specific transcriptomic signature [92]. Rajaei et al. compared the cytokine production after 1,25-(OH)_2_D treatment of whole endometrial cells and endometrial stromal cells from women with RIF and healthy fertile controls [51]. 1,25-(OH)_2_D reduced production of IL-10, TGF-β, IFN-γ, IL-6 and IL-17 and increased IL-8 levels in whole endometrial cells. In endometrial stromal cells a similar trend was observed except for an up-regulation of TGF-β in RIF patients. Endometrial cells from healthy and RIF patients produced comparable levels of vitamin D which underlines the importance of adequate circulating concentrations.

### 4.3. ART Outcomes 

Assisted reproductive technology (ART) is a term used to describe the various procedures that use the manipulation of oocytes and sperm to achieve a pregnancy. Procedures of ART include intrauterine insemination, in vitro fertilization - IVF, intracytoplasmic sperm injection (ICSI) and freezing of embryos. Since the end of the 1970s, ART has been increasingly used by many couples with an unfulfilled wish to have children to initiate a successful pregnancy. With infertility affecting over 80 million people worldwide [93] a total of around 8 million children were born worldwide following ART until 2019 [94]. Studies from the ART field suggest an impact of vitamin D status on fertility and ART outcomes.

Vitamin D deficiency is prevalent among infertile women [95] and assumed to contribute to impaired fertility [96,97] although studies that link ART outcome to vitamin D levels have revealed inconsistent results. Various studies have shown that vitamin D sufficiency is associated with an increase in pregnancy rates and live births [27,96,98,99,100]. In a retrospective cohort study of 188 infertile women undergoing IVF vitamin D deficiency was related to lower pregnancy rates in non-Hispanic white patients, but this effect was not observed in vitamin D sufficient Asian women [98]. However, controversial studies of Aleyasin et al. [2] and Firouzabadi et al. [97] propose that vitamin D status (Firouzabadi et al.: deficient (<10 ng/mL), insufficient (10–29 ng/mL), sufficient (30–100 ng/mL)) had no effect on ART outcome. Aleyasin and co-workers showed that in a study population of 101 women who have undergone ICSI those participants with sufficient follicular fluid 25-(OH)D levels had a lower embryo quality compared to women with insufficient or deficient follicular fluid 25(OH)D levels [2]. A positive correlation was found between serum and follicular 25-(OH)D levels, which indicates that follicular vitamin D level is reflective of body stores of vitamin D. 

In a systematic review and meta-analysis of 11 cohort studies including 2,700 women the associations between vitamin D levels and ART outcomes has been analyzed by Chu et al. [27]. In all 11 included studies the clinical pregnancy rate was reported as a main outcome, whereas in seven studies with 2,026 patients the live birth rate was additionally reported. A clinical pregnancy (OR 1.46, 95% CI 1.05–2.02) and live birth (OR 1.33, 95% CI 1.08–1.65) was more likely in women with sufficient vitamin D status than in deficient or insufficient women. On the other hand in six studies involving a total of 1,635 patients, no difference in miscarriage rate in women with sufficient, deficient or insufficient vitamin D concentrations following ART was found (OR 1.12, 95% CI 0.81–1.54). In a retrospective cross-sectional study of 157 women it was investigated whether poor ovarian response (POR) is associated with serum levels of vitamin D and pro-inflammatory immune reactions in infertile women with previous IVF and embryo transfer failures [2]. Women with POR and low vitamin D concentrations (<30 ng/mL) showed increased levels of CD19+B and CD19+CD5+ B-1 cells and increased TNF-α/IL-10 and IFN-γ/IL-10 producing Th cell ratios compared to women with normal ovarian response and/or normal vitamin D status. Furthermore, CD56+ NK cell concentrations (%) and NK cytotoxicity in peripheral blood were higher in women with POR and low vitamin D status compared to the other groups [70]. Given the overall positive effect of vitamin D on ART outcomes, the treatment of vitamin D deficiency could provide an important additional treatment option for many infertile women. However, there is a great lack of further large cohort and randomized studies along all ethnic groups that investigate the association of ART outcomes, the immune system and vitamin D levels.

## 5. Pregnancy Complications

### 5.1. Preeclampsia

Preeclampsia is a hypertensive disorder of pregnancy that presents in the second half of gestation with the clinical picture of new onset hypertension accompanied by proteinuria or the onset of other evidence of end organ damage [101]. In many women which ultimately develop preeclampsia invasion of cytotrophoblast and extravillous trophoblast of the maternal decidua in the first trimester of pregnancy is disturbed contributing to abnormal placentation. Several lines of evidence suggest a contribution of a dysregulated immune system in the impaired interaction between trophoblasts and decidual stroma. Components of the innate and adaptive immune system may participate in the disease development with some of them being regulated by vitamin D. 

While a Th2 dominance is a feature of healthy pregnancies [102], in preeclampsia Th1 cells dominate and the ratio of Th1/Th2 cells is increased [102,103]. A disbalance of Th1 and Th2 cells contributes to the increased release of Th1 cytokines, e.g. TNF-α and IL-6 [103,104,105,106,107,108] so that the robust inflammatory and physiologic response of pregnancies is further enhanced in preeclampsia [109,110]. High local concentrations of cytokines are involved in deficient placentation, leading to restricted proliferation, migration and invasion of trophoblasts [111,112]. Inflammatory cytokines are also capable of eliciting endothelial cell dysfunction [113] and may contribute to the elicitation of the detrimental effects on the maternal systemic endothelium that most likely mediate the disease manifestations of preeclampsia. 

Seasonal patterns in preeclampsia suggest a role for vitamin D and sunlight, because of a higher incidence of the disease in winter and a lower incidence in summer [114,115]. Compared with normal pregnancies, preeclampsia is characterized by marked changes in vitamin D and calcium metabolism [116]. Systematic reviews and meta-analyses and several observational studies suggest that low maternal serum 25-(OH)D concentrations are associated with a higher preeclampsia risk [117,118,119,120,121]. Vitamin D deficiency in pregnancy <50 nmol/L was associated with an almost 4-fold odds of severe preeclampsia [122] and vitamin D deficiency <37.5 nmol/L was even associated with a 5-fold risk of developing the disease [117]. 25-(OH)D levels have been shown to be even lower among early onset preeclamptic patients with small-for-gestational age infants compared to those preeclamptic women with adequate fetal growth, suggesting that vitamin D may impact fetal growth through placental mechanisms [123,124]. 

However, not all of the data regarding vitamin D status and preeclampsia prevalence are consistent [125,126,127]. A nested-case control study conducted in first trimester found that total and free 25-(OH)D levels were not independently associated with subsequent preeclampsia [126]. In a prospective cohort of 221 Canadian high-risk women no difference in preeclampsia prevalence was observed related to 25-(OH)D concentration [127]. Of note, the available studies are conducted in different populations and differ in their experimental set-up, definition of vitamin D deficiency, inclusion criteria and possible confounders. 

While numerous clinical and experimental studies indicate a benefit of vitamin D supplementation for preeclampsia prevention there are only few well-designed clinical trials which suggest a reduction of preeclampsia risk with vitamin D supplementation. In a controlled trial in London in the 1940’s –1950’s including 5,644 women a reduction of 31.5% in preeclampsia incidence was seen in women who received a dietary supplement containing vitamins (2,500 IU vitamin D), minerals and fish oil in comparison to the control group who did not receive any supplement [128]. Haugen et al. reported a 27% reduction in the risk of preeclampsia in a cohort of 23,423 nulliparous women in Norway who took 400–600 IU vitamin D supplements per day compared to women without supplementation [129]. However, the authors indicate that the *n*-3 fatty acid intake levels could have contributed to the potential benefit [129]. In a small RCT performed in Iran women at high-risk due to a history of preeclampsia were randomized to either placebo (*N* = 72) or 50,000 IU of vitamin D (*N* = 72) every two weeks until 36 weeks gestation [130]. The probability to develop preeclampsia was 1.94 times higher in the control compared to the vitamin D intervention group (95% CI 1.02–3.71). In contrast, a small RCT conducted in India with 400 participants, found no association of vitamin D supplementation (1,200 IU vitamin D/d and 375 mg calcium/d) and a reduced risk for preeclampsia but a reduction in diastolic blood pressure of 8 mmHg [131]. While vitamin D supplementation of 4,000 IU/d starting at 12 to 16 weeks’ gestation has been determined in an RCT as safe and most effective in achieving sufficient circulating 25-(OH)D concentrations [132], more well-designed randomized trials are needed to confirm a benefit of supplementation for preeclampsia risk reduction.

There are several possible mechanisms how vitamin D deficiency could be involved in pathophysiologic processes that cause preeclampsia, including the regulation of maternal and placental immunological and inflammatory responses, as it has been demonstrated in experimental models [31,133]. The placenta itself expresses the VDR and 1α-hydroxylase and thus produces the active metabolite of vitamin D [134], suggesting autocrine and paracrine functions of vitamin D in the placenta. There is evidence that vitamin D regulates key target genes associated with implantation, trophoblast invasion and implantation tolerance [134]. Maternal vitamin D deficiency may alter the balance of Th1 to Th2 cells in favor of Th1 cells at the implantation site and maternal-fetal interface [135]. In this line, higher expressions of Th1 cytokines have been described in placentas of preeclamptic pregnancies [136]. In addition, placental production of vitamin D is decreased in preeclampsia compared to placentas from healthy pregnancies [137,138]. The lack of local vitamin D which modulates immune function, e.g. Th1 and Th2 cells and downregulates TNF-α, IL-6 and IFN-γ expression in the placenta might contribute to the increase of those inflammatory cytokines [63,64,65]. The maternal response to impaired placentation and reduced placental perfusion in preeclampsia may equally be affected by vitamin D status. Maternal vitamin D deficiency may contribute to the increased systemic inflammatory response that characterizes preeclampsia as well as to endothelial dysfunction through direct effects on angiogenesis gene transcription [64,139,140,141,142].

### 5.2. Preterm Birth

Preterm birth occurs before 37 weeks gestation and is a major contributor to neonatal morbidity and mortality with an estimated global prevalence of 10.6% [143]. There are several reasons for preterm birth, including intrauterine infection and inflammation. Labor is a well-coordinated process and itself involves the activation of inflammatory cascades, recruitment of immune cells into the reproductive tissues and the release of inflammatory cytokines [144]. Premature activation of these mechanisms leads to preterm labor and can result in preterm birth. 

In a systematic review and meta-analysis of 6 RCTs and 18 observational studies maternal vitamin D deficiency was associated with an increased risk of preterm birth while vitamin D supplementation reduced the risk by 43% (pooled RR 0.57; 95% CI 0.36–0.91) [145]. Two vitamin D supplementation studies performed in South Carolina and a post-hoc analysis suggest that improved vitamin D status (deficient (≤20 ng/mL), insufficient (>20 – <40 ng/mL), sufficient (≥ 40 ng/mL)) in pregnancy goes along with improved health outcomes and reduced preterm birth risk [146,147]. Achieving a 25-(OH)D serum concentration ≥40 ng/ml significantly decreased the risk of preterm birth compared to ≤20 ng/mL. In contrast to these data, in a Cochrane meta-analysis of 7 trials involving 1640 pregnant women a beneficial role of vitamin D supplementation on preterm birth risk compared to no intervention or placebo wasn’t demonstrated (RR 0.66, 95% CI 0.34–1.30) [148].

Nevertheless, it seems biological plausible that a sufficient vitamin D status might help prevent preterm birth. Activation of T cells at the maternal-fetal interface and the cervix contributes to the proinflammatory response during preterm labor. Vitamin D may be important in suppressing the maternal immune response specifically Th1 cell mediated inflammatory reactions. A sufficient vitamin D status might also reduce the risk of preterm birth by maintaining myometrial quiescence. Myometrial contractility is dependent on calcium release within the muscle cell and this process is regulated by vitamin D [149,150]. Bacterial infection is one trigger of preterm birth. Laboratory studies have demonstrated links between maternal vitamin D status and placental antibacterial responses [31,37,38]. As mentioned above vitamin D is a potent inducer of the antimicrobial protein cathelicidin in the placenta and may act to potentiate placental innate immune responses [151,152].

There is biological plausibility for a protective effect of sufficient maternal vitamin D levels in pregnancy on preterm birth risk supported by results of observational and randomized studies. However, there is still a need for future research to focus on well-designed larger randomized trials and exploration of mechanisms of how vitamin D impacts physiologic processes.

## 6. Conclusions 

In recent years, the role of vitamin D in human physiology has been redefined. The effects of vitamin D are no longer based solely on calcium homeostasis and bone health, but have been extended to include its role as an immunomodulator and in the female reproductive system. 

It is known that cells of the immune system are controlled by vitamin D and that vitamin D synthesis takes place in reproductive tissues. Additionally, vitamin D plays an important role in fertility, embryo implantation and maintenance of pregnancy. Studies have demonstrated that vitamin D status affects the probability for RPL and RIF, but there is a lack of randomized clinical trials that extend the triage of vitamin efficiency, its supplementation and RPL or RIF. There are also controversial data on the effects of vitamin D deficiency on ART outcomes with some of these studies demonstrating an association of vitamin D deficiency with poorer treatment success, but this is in contrast to other studies that show that vitamin D deficiency has no impact on ART outcome. Furthermore, an association between vitamin D and pregnancy complications, e.g. preeclampsia prevalence, has also been demonstrated, but here also the data are not consistent, probably due to the different study populations and designs. 

In the present review we aimed to summarize current knowledge on the role of vitamin D in regulating immune function in reproductive tissues and outcomes. Data from the experimental and observational studies support a link between vitamin D and regulation of the innate and adaptive immune system. The few available in vitro and in vitro studies suggest in particular a modulation of the Th cell system by vitamin D status and associations to reproductive outcomes. However, the available evidence is insufficient to draw any conclusion on which vitamin D level should be aimed for to benefit from immune system modulation and achieve favorable reproductive outcomes with regards to fertility, implantation and pregnancy. Therefore, future well-designed clinical studies are needed to confirm a causal relationship of vitamin D deficiency and adverse modulation of the immune system, the effect of vitamin D supplementation and an improvement of reproductive outcomes in humans.

## Figures and Tables

**Figure 1 nutrients-12-01432-f001:**
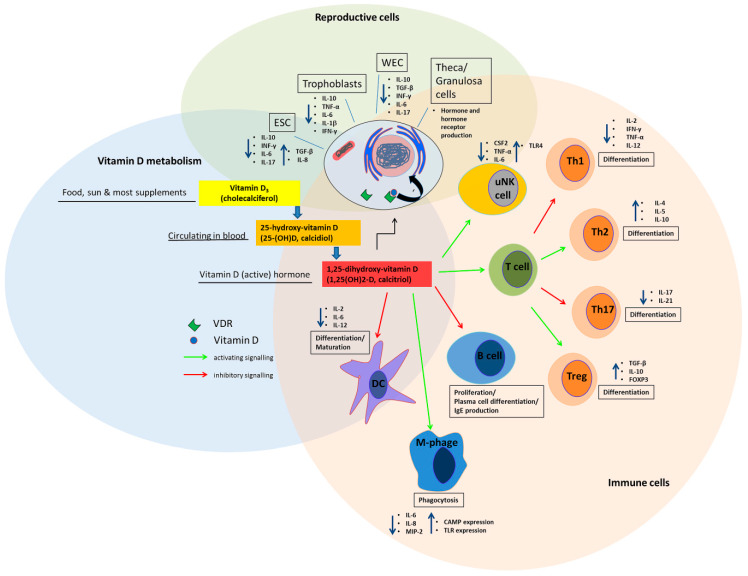
**Vitamin D effects on immune cells and cells of the reproductive tract**. Vitamin D precursors are ingested through food or supplements and further metabolized in the body to the active hormone, which exerts different responses of mediators of the immune system. Vitamin D affects maturation, differentiation, interleukin expression and immunomodulatory functions of immune cells like B cells, T cells, Th (helper) cells, Treg (regulatory) cells, macrophages (M-phage) and dendritic cells. The expression of immunoactive cytokines by cells of the reproductive tract like trophoblasts is modulated by vitamin D. Vitamin D regulates hormone (e.g. progesterone, AMH and androstenedione) and FSH and AMH receptor expression in theca and granulosa cells [4]. ESC; endometrial stem cells; uNK uterine natural killer cells; DC dendritic cell, M-Phage macrophage; Th T helper; CAMP cathelicidin, antimicrobial peptide; WEC whole endometrium cells; AMH anti mullerian hormone; FSH follicle stimulating hormone.

**Figure 2 nutrients-12-01432-f002:**
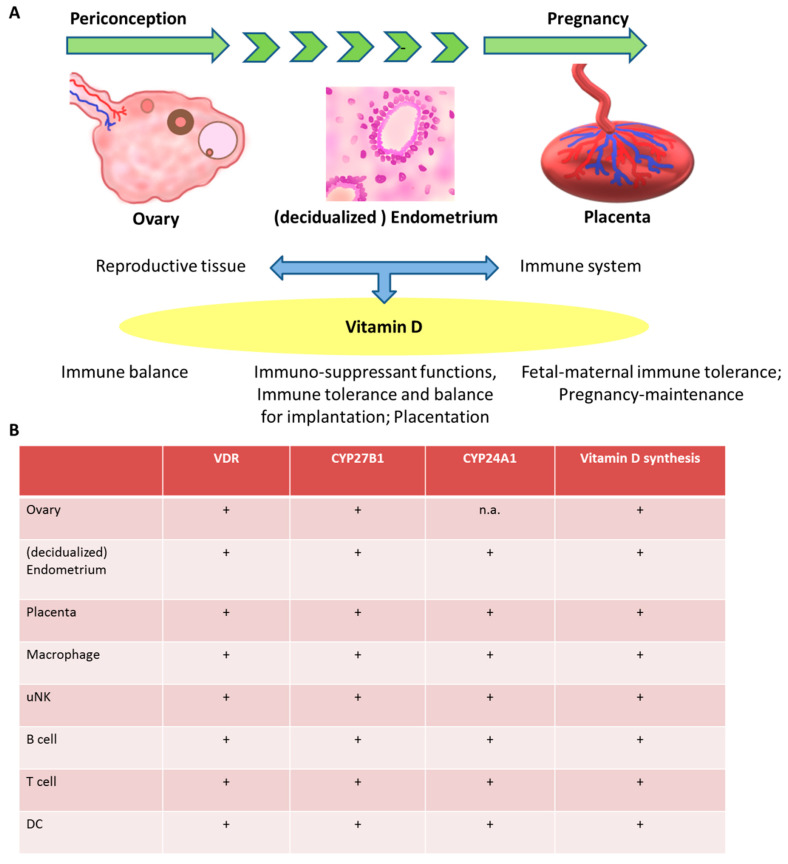
**Components of vitamin D synthesis in female reproductive tissues and cells of the immune system.** (**A**) Vitamin D exerts immunomodulatory effects with impact on reproductive tissues from the periconception period throughout pregnancy. (**B**) Components of the vitamin D metabolic system are expressed in female reproductive tissues, including ovary, decidua, endometrium and placenta and multiple immune cells. Proposed immunomodulatory effects of vitamin D include the improvement of immune balance, tolerance and maintenance of pregnancy trough effects on B cells, T cells, macrophages and dendritic cells. +; expression known, n.a.; not available.

**Table 1 nutrients-12-01432-t001:** Overview of studies focusing on the immunomodulatory effects of vitamin D in reproductive tissues and reproductive outcomes.

Reference	Main Findings
In vivo studies	
[31]	Compared to wild-type mouse placentas, placentas of VDR and CYP27B1 knock-out mice show enhanced proinflammatory cytokine and chemokine expression.
[39]	Cholecalciferol supplementation reduces the rate of LPS-induced abortions in mice. Additionally, cholecalciferol inhibits immunological modulations induced by LPS.
[61]	Vitamin D deficient mice challenged with LPS in pregnancy have higher IP-10, MCP-1, SAP, TIMP-1, VCAM-1, vWF and lower GCP-2 levels than vitamin D sufficient mice.
In vitro studies	
[22]	IL-1β induces CYP27B1 mRNA expression in human decidua cells.
[37]	Decidual NK cells treated with 1,25-(OH)_2_D or 25(OH)D synthesize less cytokines, but more CAMP.
[38]	In human trophoblasts, 1,25-(OH)_2_D induces CAMP expression. A 3A trophoblast cell line treated with 1,25-(OH)_2_D shows decreased colony forming units, when infected with E. coli.
[50]	After 1,25-(OH)_2_D treatment, cytokine expression in WECs from patients with unexplained RPL are reduced and shifted toward a Th2 phenotype. In ESCs, cytokine production is overall down-reguated, but TGF-β production is stimulated.
[51]	In WECs from RIF and normal patients, 1,25-(OH)_2_D reduces most cytokine production, whereas IL-8 is elevated. In ESCs, similar 1,25-(OH)_2_D effects are observed, except for an up-regulation of TGF-β in the RIF group.
[62]	In women with RPL, 1,25-(OH)_2_D has immune regulatory effects on NK cell cytotoxicity, cytokine secretion, degranulation process and TLR4 expression.
[63]	1,25-(OH)_2_D reduces IL-10 production in trophoblasts from normal and preeclamptic placentas.
[64]	TNF-α-induced immune response and cytokine production in human trophoblasts is inhibited by 11,25-(OH)_2_D.
[65]	TNF-α and IL-6 secretion and mRNA expression in human trophoblasts are reduced by 1,25-(OH)_2_D.
Observational studies	
[32]	Treg cell population is lower in maternal blood and cord blood in 25(OH)D_3_ deficient pregnant women. CD23 and CD21 B cell population is higher in maternal blood and cord blood in 25(OH)D deficient pregnant women. TGF-β and IL-10 levels are lower.
[35]	Maternal serum 25(OH)D deficiency is associated with placental inflammation.
[66]	Women with RPL have a lower level of CYP27B1 expression in chorionic villi and decidua compared with normal pregnant women. CYP27B1 and cytokine expression (IL-10, IFN-γ, TNF-α and IL-2) co-localize in chorionic villi and decidua cells.
[67]	In women with RPL, low 25(OH)D_3_ levels are associated with abnormalities in cellular immunity and cytokine production.
[68]	Decidual tissues of patients with RPL show less 25(OH)D, TGF-β and VDR expression and significant increase in IL-23 and IL-17.
[69]	Natural killer-1 shift in peripheral blood NK cells was identified in nonpregnant women with RPL and implantation failures.
[70]	In women with POR and low serum 25(OH)D, NK cell levels and cytotoxicity are higher. CD19^+^ B cell levels are higher, as well as the Th1/Th2 cell ratio.
	Interventional studies
[59]	Vitamin D supplementation in infertile women with insufficient 25(OH)D decreases Th1/Th2 ratio. In endometrial biopsies, 1,25(OH)_2_D_3_treatment reduces IFN-γ.
[71]	Higher percentages of CD19^+^ B cells and NK cytotoxicity, as well as a higher percentage of TNF-α-expressing Th cells are observed in RPL patients with low serum 25(OH)D levels and can be regulated to some extent with 1,25(OH)_2_D supplementation.
[72]	25(OH)D levels and Treg/Th17 ratios are decreased in women with RPL. Vitamin D supplementation increases Treg/Th17 ratio, VDR and CYP24A1 expression.
	RCTs
[73]	Vitamin D supplementation reduces Th17 cell population in peripheral blood from women with RM and reduces Th17/Treg ratio.
[74]	Patients with a history of RPL who recieved vitamin D reveal lower serum IFN-γ levels. The risk of miscarriage is reduced by 15% compared to untreated patients.
[75]	Vitamin D supplementation in women with unexplained RPL decreases serum IL-23 levels and reduces the frequency of miscarriages.

1,25-(OH)_2_D, 1,25-dihydroxyvitamin D_3_; 25(OH)D, 25-hydroxyvitamin D_3_; CAMP, cathelicidin antimicrobial peptide; CD, cluster of differentiation; CYP, cytochrome P450; E. coli, Eschericha coli; ESC, endometrial stromal cell; GCP, granulocyte chemotactic protein; IL, interleukin; IP, interferon-gamma induced protein; LPS, lipopolysaccharide; MCP, monocyte chemoattractant protein; NK cell, natural killer cell; mRNA, messenger ribonucleic acid; RIF, repeated implantation failure; RPL, recurrent pregnancy loss; SAP, SLAM-associated protein; TGF, transforming growth factor; TIMP, tissue inhibitor of metalloproteinases; TNF, tumor necrosis factor; IFN, interferon; POR, poor ovarian response; Th, T helper cell; TLR, Toll-like receptor; Treg, regulatory T cells; VDR, vitamin D receptor; VCAM, vascular cell adhesion protein; vWF, von-Willebrand-factor; WEC, whole endometrial cell.

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
