# Peer review of "Vitamin D Effects on the Immune System from Periconception through Pregnancy"

_nutrients, 2020, doi:10.3390/nu12051432_

Round 1

Reviewer 1 Report

The authors provide a short summary of select studies addressing vitamin D effects on the female reproductive system.

The association of vitamin D status with the prevention, development, progression or curability of diseases beyond bone health is a subject of enormous interest and controversial discussions.

The literature in the field was obviously studied thoroughly by the authors, many most recent references are given and the results mentioned might be of significant interest for the readers and/or researchers active in this field.

The main questions were accordingly addressed. The authors critically emphasize that observational studies are often controversial and of poor significance, partly due to relatively small size of cohorts and/or different design of studies.

The text is well written and easy to read, however, for full understanding reading of original references would be required. This is particularly the case because the authors use the term “vitamin D” as well for vitamin D3 as for other vitamin D3 metabolites, such as 25OHD3 and 1,25(OH)2D3. This appears sometimes somewhat confusing in the text and “vitamin D” needs to be formulated more precisely (examples: in line 214 “vitamin D” is meant vitamin D3, in lines 46, 101, 120, 204, 209 and 406 it is meant 1,25(OH)2D3, in lines 103 and 107 it is meant 25OHD3).

I would recommend to rephrase the sentences in lines 43 (“so far unknown” instead of “new”), 126/127  and 233 (“were obtained by Li et al. in decidual tissue which reported lower 25-(OH)D and TGF- concentrations,” to be replaced by “were obtained by Li et al, who reported…concentrations in decidual tissue”, 237: it is not clear if vitamin D supplementation or 1,25(OH)2D treatment might be a therapeutic option?, 381: “including preterm birth” to be replaced by “reduced preterm birth risk”, 415: “due to the design of the different studies” to be replaced by “due to the different design of the studies”

References: Vitamin d should be written in capital letters (D instead of d), Ref 10: D3 instead of d(3), Refs 44, 45 and 54:  OH instead of oh

Author Response

Dear Dr. Wang, dear Prof. Holick,

we thank you and the three reviewers for thoughtful and helpful comments and suggestions. We have addressed each of these comments and accordingly revised our manuscript. Changes to the manuscript are highlighted in the track-changes mode. We hope our revised manuscript is now acceptable for publication.

Sincerely,

Frauke von Versen-Höynck

Response to Reviewer 1:

Comment 1: The text is well written and easy to read, however, for full understanding reading of original references would be required. This is particularly the case because the authors use the term “vitamin D” as well for vitamin D3 as for other vitamin D3 metabolites, such as 25OHD3 and 1,25(OH)2D3. This appears sometimes somewhat confusing in the text and “vitamin D” needs to be formulated more precisely (examples: in line 214 “vitamin D” is meant vitamin D3, in lines 46, 101, 120, 204, 209 and 406 it is meant 1,25(OH)2D3, in lines 103 and 107 it is meant 25OHD3).

Reply 1: We agree with the reviewer that the terminology of vitamin D was somewhat confusing. We have now included a part in the introduction about the vitamin D terminology we used in our review (lines 40-50) and revised the terminology throughout the manuscript and more precisely state which metabolite was mentioned in the reference.

Comment 2: I would recommend to rephrase the sentences in lines 43 (“so far unknown” instead of “new”), 126/127  and 233 (“were obtained by Li et al. in decidual tissue which reported lower 25-(OH)D and TGF-b concentrations,” to be replaced by “were obtained by Li et al, who reported…concentrations in decidual tissue”, 237: it is not clear if vitamin D supplementation or 1,25(OH)2D treatment might be a therapeutic option?, 381: “including preterm birth” to be replaced by “reduced preterm birth risk”, 415: “due to the design of the different studies” to be replaced by “due to the different design of the studies”

Reply 2: We thank the reviewer for the helpful comments and rephrased these sentences in the manuscript as suggested in lines (new) 61; 284; 287; 439 and 475.

Comment 3: References: Vitamin d should be written in capital letters (D instead of d), Ref 10: D3 instead of d(3), Refs 44, 45 and 54:  OH instead of oh.Line: 889; 973; 975 and 994

Reply 3: We have now corrected these references for capital letters. Please see lines: 889; 973; 975 and 994.

Reviewer 2 Report

This article is a well-made review one to evaluate 25OHD and immunity.

It give a chance to consider vitamin d and immunity from periconception to pregnancy.

Author Response

Dear Dr. Wang, dear Prof. Holick,

we thank you and the three reviewers for thoughtful and helpful comments and suggestions. We have addressed each of these comments and accordingly revised our manuscript. Changes to the manuscript are highlighted in the track-changes mode. We hope our revised manuscript is now acceptable for publication.

Sincerely,

Frauke von Versen-Höynck

Response to Reviewer 2:

We thank reviewer 2 for his appreciating review.

Reviewer 3 Report

In the manuscript entitled to “Vitamin D effects on the immune system from periconception through pregnancy”, Bianca Schröder-Heurich et al. review vitamin D and pregnancy-related immune function. As shown in the title, there are 4 important factors such as vitamin D status, immune function, reproduction, and clinical outcomes. Despite interesting and meaningful theme, it doesn’t seem to be enough to review a role of vitamin D in clinical outcomes by modulating immune system. The manuscript should provide more detail, clear, and sound information based on in vitro, in vivo, and human studies including epidemiologic and intervention studies. It doens't seem to easy to understand their "take home message". Please make table or figures showing possible mechanism. Please let us cleary know by which vitamin D status influence implantation, pregnancy, and/or delivery by regulating immune system with reorganization.

Author Response

Dear Dr. Wang, dear Prof. Holick,

we thank you and the three reviewers for thoughtful and helpful comments and suggestions. We have addressed each of these comments and accordingly revised our manuscript. Changes to the manuscript are highlighted in the track-changes mode. We hope our revised manuscript is now acceptable for publication.

Sincerely,

Frauke von Versen-Höynck

Response to Reviewer 3:

Comment 1: Despite interesting and meaningful theme, it doesn’t seem to be enough to review a role of vitamin D in clinical outcomes by modulating immune system. The manuscript should provide more detail, clear, and sound information based on in vitro, in vivo, and human studies including epidemiologic and intervention studies.

Reply 1: We appreciate the comment and as suggested provide further details and information of in vitro, in vivo and human studies (epidemiologic and intervention). We combined this information in an additional table (table 1) and show information on in vivo, in vitro, observational and interventional studies. Studies with focus solely on immunomodulatory effects of vitamin D in reproduction are sparse and we hope with this review to stimulate further research in this field.

Comment 2: It doesn't seem too easy to understand their "take home message". Please make table or figures showing possible mechanism.

Reply 2: We appreciate the reviewers comment and to further clarify the current state of knowledge we modified figures 1 and 2. We added table 1 which provides an overview of the few in vivo, in vitro, observational and interventional studies focusing on vitamin D, immune function and reproductive outcomes. However, there is a clear lack of studies specifically investigating immunomodulatory mechanisms (e.g. pathways) of vitamin D.

Comment 3: Please let us clearly know by which vitamin D status influence implantation, pregnancy, and/or delivery by regulating immune system with reorganization. 

Reply 3: We agree with the reviewer that this is an important point. Unfortunately, we could not find any references in the current scientific literature that clearly states these data. For this reason, we have included this important point in our review and point out in the conclusion that studies on precisely this scientific question are of great relevance (line:479-483).

Round 2

Reviewer 3 Report

I am very satisfied with the revised manuscript illustrating the type of vitamin D, vitamin D status, immune function and clinical outcomes. However, line 140, 141, and 218 need to be corrected.

(line140) Cells of the reproductive tract like trophoblasts are immune modulated by vitamin D. What does this mean?

(line141) Vitamin D regulates hormone and FSH and AMH receptor expression in theca and granulosa cells. Which hormone is modulated by vitamin D?

(line 218) Please add table legend and the meaning of +.

Author Response

Dear Dr. Wang, dear Prof. Holick,

we thank reviewer 3 again for additional comments and suggestions. We have addressed each of these comments and accordingly revised our manuscript. Changes to the manuscript are highlighted in the track-changes mode. We hope our revised manuscript is now acceptable for publication.

Sincerely,

Frauke von Versen-Höynck

Reviewer 3:

Comment 1: (line140) Cells of the reproductive tract like trophoblasts are immune modulated by vitamin D. What does this mean?

Reply: We changed the wording of this sentence to “The expression of immunoactive cytokines by cells of the reproductive tract like trophoblasts is modulated by vitamin D.“ (now line 79/80)

Comment 2: (line141) Vitamin D regulates hormone and FSH and AMH receptor expression in theca and granulosa cells. Which hormone is modulated by vitamin D? 

Reply: We have added the information about vitamin influence on progesterone, AMH and androstenedione in line 81.

Comment 3: (line 218) Please add table legend and the meaning of +. 

Reply: As suggested we adjusted the table legend and added the meaning of + (now line 131).